# Interlaboratory Evaluation of a User-Friendly Benchtop Mass Spectrometer for Multiple-Attribute Monitoring Studies of a Monoclonal Antibody

**DOI:** 10.3390/molecules28062855

**Published:** 2023-03-22

**Authors:** Claire I. Butré, Valentina D’Atri, Hélène Diemer, Olivier Colas, Elsa Wagner, Alain Beck, Sarah Cianferani, Davy Guillarme, Arnaud Delobel

**Affiliations:** 1Quality Assistance sa, Technoparc de Thudinie 2, 6536 Thuin, Belgium; 2School of Pharmaceutical Sciences, University of Geneva, CMU—Rue Michel Servet 1, 1211 Geneva, Switzerland; 3Institute of Pharmaceutical Sciences of Western Switzerland, University of Geneva, CMU—Rue Michel Servet 1, 1211 Geneva, Switzerland; 4Laboratoire de Spectrométrie de Masse BioOrganique, Université de Strasbourg, CNRS, IPHC UMR 7178, 67000 Strasbourg, France; 5Infrastructure Nationale de Protéomique ProFI—FR2048, 67087 Strasbourg, France; 6Biologics CMC and Developability, IRPF, Centre d’Immunologie Pierre Fabre, 5 Avenue Napoleon III, 74160 Saint-Julien en Genevois, France

**Keywords:** multi-attribute method, monoclonal antibody, interlaboratory study, LC–MS, peptide mapping

## Abstract

In the quest to market increasingly safer and more potent biotherapeutic proteins, the concept of the multi-attribute method (MAM) has emerged from biopharmaceutical companies to boost the quality-by-design process development. MAM strategies rely on state-of-the-art analytical workflows based on liquid chromatography coupled to mass spectrometry (LC–MS) to identify and quantify a selected series of critical quality attributes (CQA) in a single assay. Here, we aimed at evaluating the repeatability and robustness of a benchtop LC–MS platform along with bioinformatics data treatment pipelines for peptide mapping-based MAM studies using standardized LC–MS methods, with the objective to benchmark MAM methods across laboratories, taking nivolumab as a case study. Our results evidence strong interlaboratory consistency across LC–MS platforms for all CQAs (i.e., deamidation, oxidation, lysine clipping and glycosylation). In addition, our work uniquely highlights the crucial role of bioinformatics postprocessing in MAM studies, especially for low-abundant species quantification. Altogether, we believe that MAM has fostered the development of routine, robust, easy-to-use LC–MS platforms for high-throughput determination of major CQAs in a regulated environment.

## 1. Introduction

The multi-attribute method (MAM) by mass spectrometry (MS) is well established across the industry in non-Good Manufacturing Practice (GMP) environments for antibody-based product and process characterization purposes [1,2,3,4]. Most pharmaceutical companies and many instrument providers are currently working on the extension of MAM to quality control (QC) labs, as initially proposed by Amgen in 2015 [5]. Targeted monitoring of critical quality attributes and “new peak detection” (NPD) are required to establish MAM by LC–MS peptide mapping as a purity assay in a QC environment [6]. MAM by LC–MS peptide mapping has the proven capability to replace multiple conventional HPLC/CE-based QC methods such as charge variants (ion-exchange chromatography—IEX, capillary isoelectric focusing—cIEF, capillary zone electrophoresis—CZE) [7], fragments (reduced capillary electrophoresis–sodium dodecyl sulfate—rCE-SDS), oxidation (reversed-phase chromatography—RPC, hydrophobic interaction chromatography—HIC, peptide mapping, LC-UV), N-glycans [8,9], sequence variants (LC–MS/MS) and identity tests (ELISA, peptide mapping, LC-UV) [10]. The technology is well-advanced, with instruments and software solutions being developed by several vendors allowing routine use in a GMP environment [11].

The implementation of MAM is supported by established guidelines (e.g., ICH Q2, ICH Q6B, ICH Q14) and could facilitate advanced control strategies in line with ICH Q8 [12]. The expected benefits are improved quality control testing and shortened development timelines through enhanced product and process understanding. The aim is also to provide quantitative information on individual site-specific critical quality attributes (CQAs), thereby enabling more specific control of the safety and efficacy of the drug. The increase in speed by leveraging MAM as the platform method with the potential for automation is another advantage [13]. MAM should also help to de-risk accelerated development by retrospective assessment of newly identified product quality attributes (PQAs) using previous data sets.

The use of MAM for lot release and stability testing according to GMP is not well established across the industry, due to the ongoing evolution and alignment of best practices, the complexity of the method (sample preparation, instrumentation, data analysis), as well as limited experience and regulatory unfamiliarity with the filing of MAM as a QC tool [14,15]. Replacing conventional methods within diverse and unclear regulatory landscapes can be seen as a potential business risk. This could be improved by increased effort and risk management through parallel testing using MAM and conventional methods. The limited experience validating new peak detection (NPD) could be addressed by interlaboratory studies and step-by-step appropriate specifications setting [6,16].

The typical limitations of MAM by LC–MS peptide mapping include the detection of clipping sites (degradation), the presence of small peptides that cannot be efficiently retained on the LC column, the potential risk of sample preparation-induced artifacts including peptide modification and missed cleavages and chromatographic tailing or carry-over of hydrophobic peptides [17]. Moreover, the bottom-up approach that is widely used can make it difficult to link the results obtained at the peptide level to what occurs on the intact protein.

Here we report an interlaboratory study of MAM for nivolumab. Nivolumab is a gold standard immune-oncology mAb directed against PD-1 as a single agent, or in combination treatments, in advanced, unresectable or metastatic melanoma in adults [18]. It is a hIgG4k antibody, stabilized in the hinge region by an S228P mutation to avoid half antibodies [19,20], and produced in CHO cells with the representative and well-known N-glycans [21,22].

It was recently approved by the FDA as a combination treatment for adult and pediatric patients with unresectable or metastatic melanoma with relatlimab, an anti-LAG-3 antibody at a fixed dosage (Opdualag; 160 mg relatlimab and 480 mg nivolumab) [23]. Such co-formulated antibody-based drug products are more and more frequent and could benefit from MAM for batch release and stability studies.

In this study, we aimed at evaluating the repeatability and robustness of a compact, easy-to-use benchtop LC–MS system along with its corresponding data treatment pipeline for the analytical characterization of monoclonal antibodies (mAbs). The same samples were thus distributed to three independent laboratories equipped with the same LC–MS instrumentation, using the same columns and chromatographic gradients, with the aim of evaluating CQAs such as asparagine deamidation, methionine oxidation, C-terminal lysine deletion (K-clipping) and glycosylation, on three types of nivolumab samples. The performance of the MAM method was evaluated by comparing the non-stressed (“control”) to either pH-stressed or oxidative-stressed samples. As the data processing is suspected to potentially have a significant impact on the results, various software solutions were considered. Two of those solutions (UNIFI and waters_connect) are compliant with regulatory requirements in terms of data integrity (including 21 CFR Part 11) [24,25], and an open-source solution (Skyline) was tested in parallel. UNIFI can be used for the process of all MS peptide mapping data, while waters_connect contains an additional application, Peptide MAM, specifically dedicated to MAM processing. This adds more performance and flexibility to the process. Skyline is open-source software with full flexibility and control over the process, but it cannot be used in a regulated environment. Such a comparison of different data treatment procedures has never been reported before in the context of peptide MAM.

## 2. Results and Discussion

### 2.1. Study Design

The peptide mapping of a reference mAb product was conducted in three different laboratories equipped with strictly similar benchtop LC–MS equipment. In the present study, the different samples were prepared in one single laboratory, and aliquots of the digests were shipped to the other participants. As a result, any observed variability could not be attributed to the tryptic digestion procedure, which is known to be one of the potential sources of variability in peptide MAM workflows [17]. In addition, the data treatment was also performed in one unique laboratory having a high degree of expertise, and various software solutions were considered to evaluate its impact, including two 21 CFR Part 11-compliant software solutions, UNIFI and waters_connect, and an open-source solution, Skyline.

Nivolumab was selected as the model biopharmaceutical product to check the applicability of the peptide MAM strategy. Appendix A shows the amino acid sequence of the nivolumab light chain (LC) and heavy chain (HC). When using the developed LC–HRMS method, a sequence coverage of 100% was achieved when considering an MS error of 10 ppm and no constraints on MS/MS fragments, and a good sequence coverage of 93% was achieved when including at least two MS/MS fragments to confirm the identification. This confirmed the correct enzymatic digestion of the sample and validated the use of this methodology for the analysis of nivolumab in this study. Two stressed conditions were applied to the nivolumab sample (high pH as stress #1 and oxidative stress with H_2_O_2_ as stress #2). The three different samples (control, stress #1 and stress #2) were then injected in triplicate in three different laboratories. 

The representative total ion chromatograms (TICs) of the control, stress #1, and stress #2 samples are reported in Figure 1. As shown, there is a significant number of peptides that can be chromatographically resolved using the developed generic LC conditions. When comparing the control and pH stress samples, the differences were quite limited, with only a very few minor changes at the chromatographic level. However, the oxidative stress sample was found to be much more dissimilar from the control sample, with both the apparition of new peaks and changes in peak intensities.

Seven peptides (Appendix A) subjected to post-translational modifications (PTMs) were monitored during the MAM study. The retention times of these peptides and their modified versions were reported in Appendix A. Retention times do not vary significantly when performing three replicate injections on one given instrument (RSD values on retention times were, on average, equal to 0.08% and ranged from 0.02 to 0.35% in the worst case). However, when combining the retention times obtained in the three different laboratories, the RSD values were, on average, equal to 1.4%, with values ranging from 1.1 to 1.9%. As the reported values were always lower than 2%, we hereby prove that the chromatographic separation was reproducible across the laboratories and that its variability would not impact the interpretation of the study results. The minor differences observed between the three laboratories were probably due to differences between the LC hardware (although the same models were used, there were differences in terms of tubing or mixers) and C18 columns (they did not have the same history in terms of the number of injections and type of samples analyzed). These minor differences could occur during a real MAM analysis in QC analyses and are therefore relevant to evaluate the possibility to implement the MAM methodology in a biopharmaceutical environment.

Several well-known critical quality attributes (CQAs) arising from protein bioproduction and degradation were tracked and quantified in this work, including deamidation, oxidation, C-terminal lysine truncation and glycosylation. These chemical modifications are highlighted in Appendix A and further described in the next sections. 

### 2.2. Asparagine Deamidation

Asparagine deamidation is a non-enzymatic modification of asparagine (N) and, to a lesser extent, glutamine (Q), which occurs via a loss of NH_3_, formation of a cyclic imide (succinimide) intermediate and hydrolysis, leading to the formation of aspartic (D) and iso-aspartic (iso-D) acids [26]. It is a common chemical modification of mAb that can result in loss of antigen binding and bioactivity [21]. Deamidation leads to a mass change of +0.984 Da, along with a reduction of the isoelectric point (pI), due to the change in charge of one amino acid from neutral to negative [27]. Interestingly, the two reaction products of deamidation can be separated for the main peptide by reversed-phase liquid chromatography (RPLC), and deamidation can be accelerated by increasing either pH or temperature.

In the present work, nivolumab was exposed to high pH stress (pH 9 at 40 °C) to increase the level of deamidation (forced degradation study [28]). As expected, the most common deamidation site of nivolumab was located in the PENNY peptide of the Fc domain, namely the peptide HC (364–385) having sequence GFYPSDIAVEWESNGQPENNYK.

Figure 2 highlights the percentage of deamidation observed on the PENNYK peptide in the three different laboratories for the high pH-stressed sample and using the three different types of software for the data treatment. As shown, the amount of deamidation was always low, with values between 5 and 7%. Interestingly, these values do not vary too much, whatever the instrument on which the experiments were performed and the software on which the data treatment was done.

From a chromatographic point of view, the PENNYK peptide was eluted with a retention time of 34.50 min. When extracting the *m/z* ratio corresponding to this peptide (charge state +2), three additional peaks were observed at close retention times of 34.30, 34.74 and 34.88 min (see Appendix A). Based on the fragmentation spectra, these three peaks correspond to the various deamidated versions of the native peptide, which cannot be resolved with our MS instrument offering a resolution of only ~10,000. Interestingly, the PENNYK peptide possesses three asparagine (N) residues susceptible to deamidation. The two peaks eluted after the main peptide were identified as the deamidation of the two asparagine in the PENNY section of the peptide, while the deamidation of the NG section of the peptide produces the peak eluted before the GFYPSDIAVEWESNGQPENNYK peptide. In Figure 2, the percentage deamidation results were reported only for the peak eluted at 34.30 min.

The resolution between these four peaks was found to be insufficient (resolution was between 0.8 and 1.2), because of similar hydrophobicity and overlapping mass spectra. This limited resolution makes the integration of the peak of interest (eluted at 34.30 min) difficult. Depending on the way the integration is performed by the software (the software employed in this work can either perform an automated integration of peaks based on settings entered by the user (UNIFI and waters_connect), or a manual integration set by the users (Skyline)), the peak areas of the native and deamidated peptides can vary, leading to differences in the percentage of deamidation, as highlighted in Figure 2. In addition, some minor differences in terms of C18 column selectivity and the LC pumping system between the three different instruments can also be responsible for the slight changes in selectivity/resolution observed between the different instruments (see Figure 2). 

Nevertheless, the observed differences in terms of the percentage of deamidation were found to be fully comparable, whatever the employed instrument and software, using the peptide MAM approach (Appendix A).

### 2.3. Presence of C-Terminal Lysine

The presence of C-terminal lysine on the heavy chain (HC) is another common CQA that reflects the manufacturing consistency. C-terminal lysine residue is typically removed from the antibody during the production process due to carboxypeptidase activity [29]. This modification is easy to visualize with MS, as it generates a mass shift of +128 Da (variant with C-terminal lysine) compared to the truncated peptide. From a chromatographic point of view, the presence of C-terminal lysine results in the formation of a less-hydrophobic variant that can also be easily separated by RPLC.

Figure 3 shows the amount of the C-terminal lysine variant for the different instruments and data treatment protocols for the control sample (K-clipping is not expected to be impacted by pH or oxidative stress). In all cases, the percentage of the C-terminal lysine variant was very low and comparable, with an average amount of 2.1%, and values ranging from 1.5 to 2.5%. Interestingly, no statistical difference was observed between the three instruments. However, it appears that one of the software solutions (UNIFI) systematically underestimates the amount of C-terminal lysine variant by about 0.7%.

To better understand this observation, the quality of the data was manually checked. At the chromatographic level, the two peaks corresponding to the SLSLSLGK and SLSLSLG peptides are easily separated (retention times of 26.79 and 29.56 min) and the S/N ratio was very good in both cases, so the integration of these two peaks was easily obtained and could not be the reason for the observed discrepancies. We next checked if the same charge states were considered for the two species. In both cases, the +1 and +2 species were considered, so this cannot explain the slight difference observed with the UNIFI software. The difference was probably due to differences between the algorithms used in UNIFI and waters_connect: in UNIFI, each injection is processed separately, without any retention time alignment between injections. In waters_connect (with Peptide MAM application), the injections’ data are first aligned based on retention times and then combined to produce a map of all the ion species in all the injections. The corresponding map is then used for peak picking, and the alignment information allows us to quantify each peak on all the injections, rather than just those injections for which there was a strong signal. This could explain why a higher average response is observed in the waters_connect for low-level species.

Despite this minor difference observed with one software solution, it appears that the presence of the C-terminal lysine variant can be reliably evaluated with the peptide MAM method developed in this work.

### 2.4. Methionine Oxidation

Recombinant mAbs are often exposed to oxidizing environments (i.e., dissolved oxygen, oxygen in the air, free radicals, etc.). The most susceptible residue to oxidation is methionine (M) and, to a lesser extent, tryptophan (W). In the presence of an oxidizing reagent, methionine residue is oxidized to create methionine sulfoxide, with a mass increase of +16 Da, and, to a lesser extent, methionine sulfone, with a mass increase of +32 Da [28]. Oxidation is considered an important CQA and is one of the main causes of concern for formulation scientists. Oxidation is indeed known to impact the structural integrity, conformational stability, safety, and efficacy of the mAb product [27]. In terms of the analytical procedure, oxidation modifies the surface hydrophobicity due to the partial unfolding of oxidized proteins, which exposes the hydrophobic amino acid residues. Therefore, the characterization and monitoring of oxidation can be easily assessed with peptide mapping via RPLC–MS. 

In this study, nivolumab was oxidized in the presence of hydrogen peroxide (H_2_O_2_) and analyzed using the generic MAM approach. Various methionine residues located on different parts of the nivolumab HC were oxidized under such conditions, and the corresponding peptides (i.e., ASGITFSNSGMHWVR, DTLMISR, GQPREPQVYTLPPSQEEMTK and NTLFLQMNSL) were quantified using the three different LC–MS instruments at our disposal. Figure 4 shows the corresponding percentage of oxidation for these four peptides with the different software solutions. As illustrated, the percentages of oxidation for the four selected peptides were between 6 and 20%. In addition, the variability between the instruments and the software was sometimes significant. For DTLMISR, the percentage of oxidation varies from 14 to 21%, while it varies from 5.7 to 9.5% for ASGITFSNSGMHWVR, from 4.4 to 9.0% for NTLFLQMNSL and, more critically, from 3.5 to 16% for GQPREPQVYTLPPSQEEMTK. In our opinion, there are three main sources of variability. 

First, the amount of methionine added to the sample at the end of the reaction was probably not sufficient to completely stop the H_2_O_2_ oxidation. As the samples were not analyzed at the same time in laboratories 1, 2 and 3, the variations in terms of the percentage of oxidation can probably be partially explained by the different storage durations. For example, this is illustrated by the peptide ASGITFSNSGMHWVR, which was first analyzed in laboratory #2, then laboratory #1 and, finally, laboratory #3. As shown, the level of oxidation was lower in laboratory #2 (5.9% oxidation on average), slightly higher in laboratory #1 (7.7% oxidation on average) and even higher in laboratory #3 (8.8% oxidation on average).

Secondly, as was reported for the deamidation, it appears that the automatic integration of the chromatographic peaks may also impact the peak areas and, therefore, the percentage of oxidation, as the peak shapes obtained in the three different laboratories were not strictly identical, and the diverse software solutions also use a different algorithm for peak integration. This was particularly true for the GQPREPQVYTLPPSQEEMTK peptide, which appears as a broad, noisy peak, and sometimes with significant tailing, thus providing important differences in terms of peak areas upon integration (data not shown). The integration issue was also observed with the DTLMISR and NTLFLQMNSL peptides but to a lesser extent. For these last two peptides, the issue was due to the presence of a contaminant peak eluted quite close to the oxidized peptide and not differentiated by MS. Interestingly, Skyline offers much more flexibility for peak integration, as the start and end of the peak integration can be directly changed on the XIC by the analyst. On the other hand, UNIFI and waters_connect only allow performing an automatic peak integration based on parameters set by the user (21 CFR Part 11-compliant software).

Finally, another source of variation could be related to an inappropriate automatic selection of charge states in the MS spectra. Indeed, in some cases, different charge states were considered depending on the software. The selected charge states can be easily modified with Skyline or waters_connect, but are automatically selected in UNIFI, with no possible modification. This can lead to issues; as an example, for the peptide GQPREPQVYTLPPSQEEMTK in laboratory #1, the charge states +2 and +3 were selected in UNIFI for the first replicate, while only the charge state +2 was selected for the two other replicates. In laboratory #2, charge state +3 was selected for the first and third replicates, while +2 was selected for the second replicate. In laboratory #3, the +3-charge state was systematically selected for the three replicates. 

In the end, the variability in terms of the percentage of oxidation between the different laboratories and software was found to be acceptable for the DTLMISR, ASGITFSNSGMHWVR and NTLFLQMNSL peptides with the peptide MAM approach. However, the variability was clearly too high for GQPREPQVYTLPPSQEEMTK, mostly due to the difficulty with peak integration, which should be corrected manually to draw reliable conclusions. It should be noted that this peptide contains three proline residues, which can induce an uncommon behavior in mass spectrometry, notably in terms of fragmentation. By selecting this kind of peptide in a MAM study, one should accept higher variability.

### 2.5. Glycosylation

Finally, we also explored the glycosylation of nivolumab using the peptide MAM approach. Glycosylation is considered one of the most important CQAs, because of its major role in immunology and impact on biological activity, mAb stability and toxicity [30]. Like the vast majority of mAbs, nivolumab is N-glycosylated at the conserved asparagine (N) residues in the CH2 domain. There is a certain degree of heterogeneity that can be observed for nivolumab in terms of glycosylation, with three major glycoforms made with a core-fucosylated structure with either zero (G0F), one (G1F) or two (G2F) galactoses. In addition, we also observed the G0 glycoform (absence of core fucose) and non-glycosylated mAb species [31]. From an analytical point of view, glycosylation can be characterized at various levels (e.g., intact mAb or sub-units, glycopeptides, release glycans or monosaccharides) [32]. Even if the most informative strategy remains the released glycan approach, we have tried to evaluate the possibility offered by the peptide mapping approach, as it has proven its applicability in the past [33].

Figure 5 shows the amounts of G0, G0F, G1F, G2F and non-glycosylated species found in nivolumab in the three different laboratories and using the various data treatment procedures for the control sample. All the values reported in Figure 5 correspond to the average of nine values (three replicates of the three samples, i.e., control, pH stress and oxidative stress). The repeatability was found to be excellent whatever the instrument and software, as the RSD values on the percentage of glycoform were between 0.1 and 1.1% of the maximum (see error bars).

As shown in Figure 5, G0F was the main glycoform (about 70%), followed by the G1F (around 13%), non-glycosylated (about 9%), G2F (about 4%) and G0 (about 1.5%). In the case of glycosylation, the variability was found to be very low using the peptide MAM approach, and all the values were not statistically different, whatever the instrument and software, for the five different glycoforms. However, it is important to mention that two partially separated peaks, eluted at 42.60 and 42.71 min, were observed for the G1F species. These two peaks are due to the presence of two isomers for G1F (depending on the position of the galactose on the glycan structure). Depending on the software, either one or two of these peaks were considered. As a result, it was important to manually verify which peak was integrated by the software. In this work, the data reported in Figure 5 corresponds to the integration of only the first peak.

Finally, the peptide MAM approach works particularly well for the quantitation of glycans, and no variability was observed between the laboratories and software.

## 3. Materials and Methods

### 3.1. Chemicals and Reagents

An AccuMap Low pH Protein Digestion Kit (VA1040) was obtained from Promega (containing Lys-C and trypsin enzymes, as well as tris(2-carboxyethyl)phosphine (TCEP), iodoacetamide, a denaturing solution and low-pH reaction buffer). Nivolumab (Opdivo 11024618-A—batch ABL0198) was purchased from Euromedex. Water (23214102) and acetonitrile (01207802) were purchased from Biosolve. All the other reagents were of analytical grade and were purchased from Merck or Sigma-Aldrich (St. Louis, MO, USA). 

### 3.2. Sample Preparation

Two types of stressed nivolumab samples were prepared for this study to induce the modifications. One aliquot was incubated at a final concentration of 2 mg/mL at pH 9 in a 50 mM Tris buffer at 40 °C for 5 days. After five days, the sample was stored at −20 °C until the enzymatic digestion. Another aliquot was incubated for 45 min at room temperature in the presence of 0.05% H_2_O_2_ at a final concentration of 2.5 mg/mL. The oxidative stress was stopped by the addition of 50 mM of methionine (final concentration) to the solution. The final concentration of nivolumab was 2 mg/mL. The digestion was performed immediately after the oxidative stress. 

The enzymatic digestion was performed with the AccuMap Kit on both the stressed samples and the control sample (nivolumab diluted at 2 mg/mL), each in triplicate. The denaturation and the reduction of the disulfide bridges were performed by adding 40 µL of denaturing solution to 25 µL of nivolumab at 2 mg/mL, followed by the addition of 2 µL of 100 mM TCEP and 7 µL of low-pH reaction buffer. The solution was homogenized and incubated for 30 min at 37 °C. After this reduction step, 4 µL of 300 mM iodoacetamide was added to all the samples for alkylation, and the solutions were incubated for 30 min at 37 °C, protected from light. A pre-digestion step with Lys-C was performed by adding 5 µL of low pH buffer and 25 µL of LysC and further diluted with 12 µL of H_2_O. This solution was incubated for 1 h at 37 °C. At the end of the pre-digestion, the digestion step was performed by adding 75 µL of H_2_O, 20 µL of low-pH buffer, 25 µL of Lys-C, 20 µL of trypsin and 60 µL of 80 mM methionine (to suppress oxidation). The samples were incubated for 3 h at 37 °C. The digestion was stopped by the addition of 32 µL of 20% TFA. The digests were pooled by condition, vortexed and frozen before shipping to the different sites for independent analysis. 

### 3.3. Generic Chromatographic Conditions

The peptide separation was performed on a Waters Acquity UPLC Peptide BEH C18 column (130 Å, 1.7 μm, 2.1 mm × 100 mm). Eluent A was 0.1% TFA in water and eluent B was 0.1% TFA in acetonitrile. Elution was performed at 65°C at a flow rate of 0.2 mL/min with a gradient as follows: linear gradient from 1% B to 42% B at 60 min, followed by a linear gradient to 80% B at 61 min and an isocratic step at 80% B for 3 min before going back to the starting conditions for equilibration (1% B).

The samples (10 µL) were injected in triplicate on each LC–MS system, and a blank injection was performed between each triplicate injection. 

### 3.4. LC–MS Instrumentation and MS Settings

In the three different laboratories, the instrument consisted of an ultra-high-performance liquid chromatography (UHPLC) system (Acquity UPLC I-Class Plus, Waters Milford, MA, USA) coupled to a time-of-flight (TOF) mass spectrometer (BioAccord Acquity RDa, Waters Milford, MA, USA). Each UHPLC system was equipped with a binary solvent delivery pump, a 50 µL mixer assembly (or a 380 µL mixer assembly in lab 2), a flow-through needle (FTN) sample manager, and a Tunable UV detector (TUV) operated in the single wavelength mode (λ = 214 nm, sampling rate 20 points/s). For all the experiments, the MS device was operated in the ESI+ mode with a capillary voltage of 1.5 kV, a desolvation temperature of 500 °C and a cone voltage of 30 V. The peptide fragmentation was obtained by ramping up the fragmentation cone voltage from 60 V to 130 V. The full scan acquisition was performed with Intelligent Data Capture (IDC) enabled and a mass range of 50–2000 *m*/*z* with a scan rate of 5 Hz. The data acquisition was performed with UNIFI Software (Waters, Milford, MA, USA): UNIFI v1.9.4 in lab 1, v1.9.9 in lab 2 and v1.9.13.9 in lab3. 

### 3.5. Data Analysis

The extent of the digestion and data quality was evaluated by first analyzing a dataset with UNIFI 1.9.4 (Waters) with the peptide mapping process. The peptide search was based on the nivolumab light and heavy chain sequences. The peptide identification was performed with a mass tolerance of 10 ppm on the precursor and 15 ppm for the fragments. Carbamidomethylation was set as a fixed modification as well as the pyroglutamic acid E N-term and Q N-term and loss of Lysine C-term. The oxidation of methionine, glycosylation (G0F, G1F, G2F, G0) and the deamidation of asparagine were selected as the variable modifications. This process allowed for the selection of peptides for the multisite analysis. The seven selected peptides and their modified counterparts were directly sent to the UNIFI library (peptide sequence, *m*/*z* and retention time). 

The percentages of modification in these different peptides were compared on three different software platforms. One process was performed in UNIFI using the Accurate Mass screening process (UNIFI 1.9.4), with the peptide list in the created library. The quantification was based on the extracted ion chromatograms, as the sum of all the adducts and isotopes. The data were expressed as a percentage modification based on the response of the extracted ion chromatograms. 

Another process was performed in the peptide MAM application (waters_connect v1.1.0.5, Waters) in which the attributes were imported from the same UNIFI library. The process was performed automatically using different charge states and the isotopes of each attribute and data were reported as a percentage modification. 

A third process was performed in the freely available and open-source Skyline application [34] (MacCoss Lab, Department of Genome Sciences, University of Washington, v.22.2.0.312). Prior to the Skyline processing, the UNIFI sample set was exported to Masslynx in .raw files. Skyline extracted the retention time and intensity information for a given *m*/*z* from the peptide list. Manual integration was performed when necessary. The total area, corresponding to the sum of the three extracted isotope areas from the different charge states, was used to calculate the percentage modification.

## 4. Conclusions

We provide here a small-sized interlaboratory study to evaluate a MAM workflow performed on benchtop LC–MS equipment. Overall, our study highlights the consistency of the peptide mapping results obtained within this study using identical benchtop LC–MS platforms and nivolumab samples prepared by one group and distributed to the other laboratories to minimize sample preparation artifacts. The post-processing and relevance of the bioinformatics data treatment were investigated by centralizing the data interpretation at one site with three different commercial or open-source pipelines, avoiding the study of the impact of parametrization of the different software packages.

We highlighted the importance of the downstream process of the analytical pipeline, i.e., the bioinformatics processing and statistical analysis, which represent critical steps to generate the final comparative results, by comparing two 21 CFR Part 11-compliant software solutions provided by the vendor and an open-source tool classically used for quantification in targeted proteomics (used as the versatile workflow reference). Our study first demonstrates that, independently of the post-processing workflow, peptide-mapping MAM provides consistent results for the glycoform analysis of the major drug product, with excellent repeatability, suggesting that glycoform analysis can be monitored by peptide-mapping MAM. For more minor proteoforms (oxidation, deamidation, N-terminal K-clipping), the performance of the software might be significantly different, mainly due to the automatic peak integration, selection of charges states or quantification algorithms, putting the selection of the bioinformatics pipeline at the forefront. More flexible tools such as Skyline offer improved possibilities for back corrections of mis-peak integration or mis-charge selection than 21 CFR part 11-compliant software, with waters_connect being more flexible than UNIFI, and thus better adapted for MAM studies. However, such an open-source solution is not applicable in a regulated biopharmaceutical environment, due to the lack of compliance with data integrity regulatory requirements. 

Altogether, our results suggest that user-friendly LC–MS equipment combined with optimized data processing pipelines support the integration of peptide mapping in MAM strategies. Peptide mapping MAM methods should include a preliminary extensive evaluation of the choice of the peptides to monitor, selection of charge states to consider and quality of the peak integration to provide optimal robust workflows. Bioinformatics post-processing is thus of utmost importance, and there is still room for improvement for more accurate automatic LC and/or MS peak integration, especially when considering low-intensity peptides. The influence of the processing algorithm on the results obtained is a point of attention for the implementation of the MAM approach in the biopharmaceutical industry: if different laboratories are to be involved in a project, several software solutions may have to be used. In this case, the comparability of the results obtained should be evaluated early in the method development process, and, if applicable, the method should be validated using the different software solutions in parallel. The use of a well-characterized reference standard such as NISTmAb [35] could also be used to facilitate the comparison across the software solutions.

It should be noted that all the samples employed in this study were prepared in the same laboratory to avoid variability due to sample preparation. To apply the MAM methodology in a QC environment, sample preparation should be carefully controlled, for instance by using automation [17]. New peak detection was not considered in this study, as it is strongly dependent on the software solution used and cannot be readily validated, nor is it easily amenable to QC testing. However, the development and validation of a robust NPD workflow are considered prerequisites for the use of MAM by LC–MS as a purity assay and the successful replacement of conventional methods.

In the future, MAM could also be used for more complex antibody-based products such as Fc-fusion proteins, bi- or multispecific antibodies or antibody-drug conjugates (ADCs) [36,37]. In addition, we expect that intact level MS-based protein analysis (intact mass measurement and/or subunit middle-level analysis) might reinforce current peptide-centric based MAM workflows, as recently proposed [38].

## Figures and Tables

**Figure 1 molecules-28-02855-f001:**
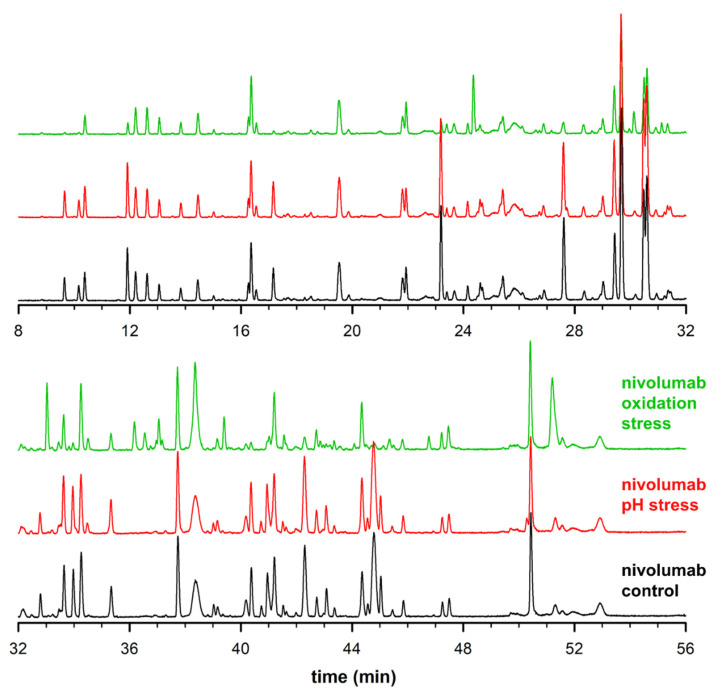
Total ion chromatograms (TICs) of control (**black**), high pH stress (**red**) and oxidative stress (**green**) samples (examples of chromatograms obtained in lab 2).

**Figure 2 molecules-28-02855-f002:**
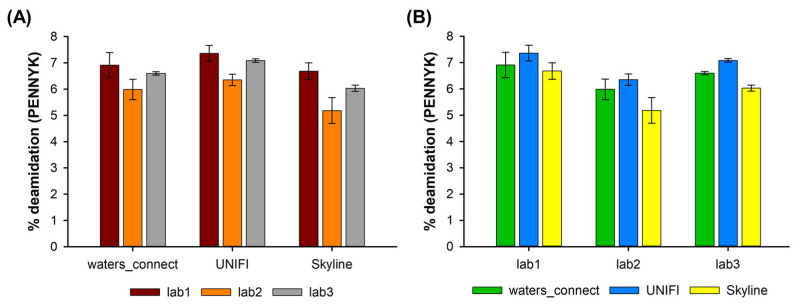
Percentage of deamidation observed on the PENNYK peptide in the high pH-stressed sample for the peak eluting at 34.30 min evaluated (**A**) by using three different software solutions (waters_connect, UNIFI and Skyline) on data obtained in (**B**) three different laboratories (lab1, lab2, and lab3).

**Figure 3 molecules-28-02855-f003:**
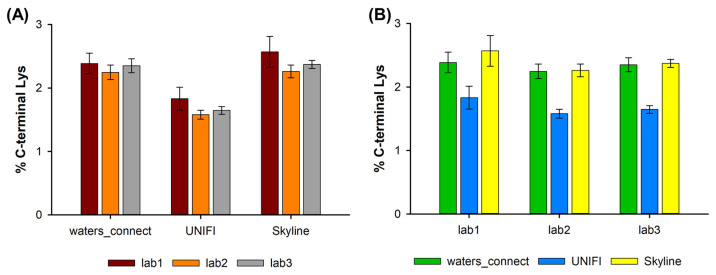
Amount of C-terminal lysine observed on the SLSLSLGK peptide evaluated (**A**) by using three different software solutions (waters_connect, UNIFI, and Skyline) on data obtained in (**B**) three different laboratories (lab1, lab2, and lab3).

**Figure 4 molecules-28-02855-f004:**
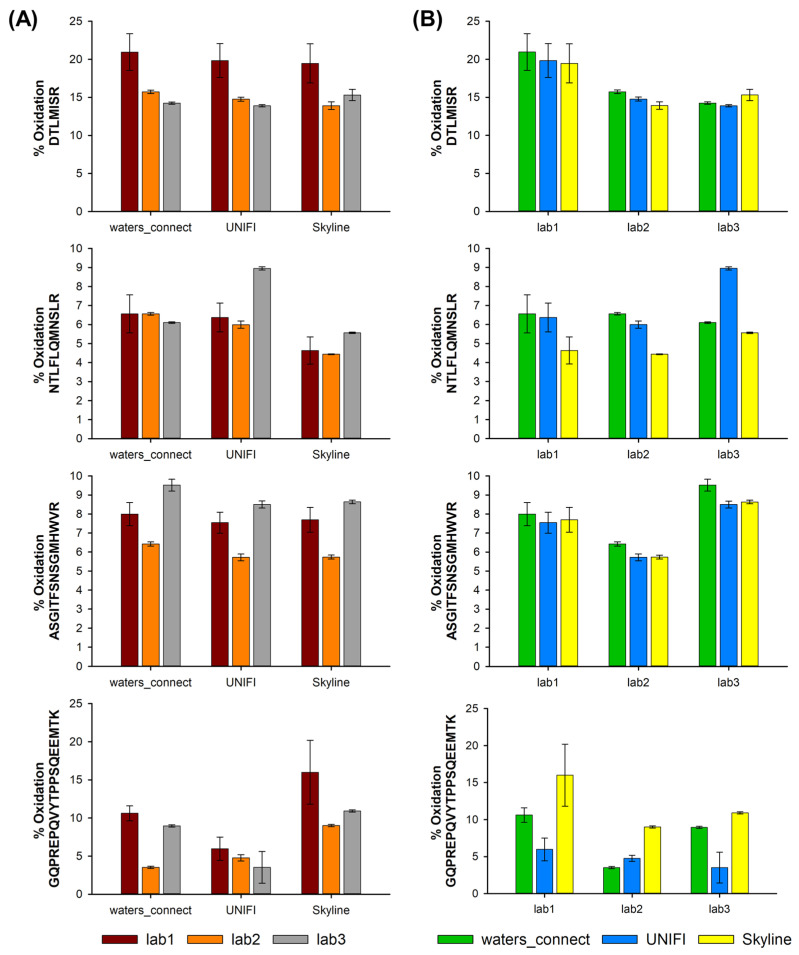
Percentage of oxidation observed on four different peptides, namely, DTLMISR, GQPREPQVYTLPPSQEEMTK, ASGITFSNSGMHWVR and NTLFLQMNSL, evaluated (**A**) by using three different software solutions (waters_connect, UNIFI and Skyline) on data obtained in (**B**) three different laboratories (lab1, lab2 and lab3).

**Figure 5 molecules-28-02855-f005:**
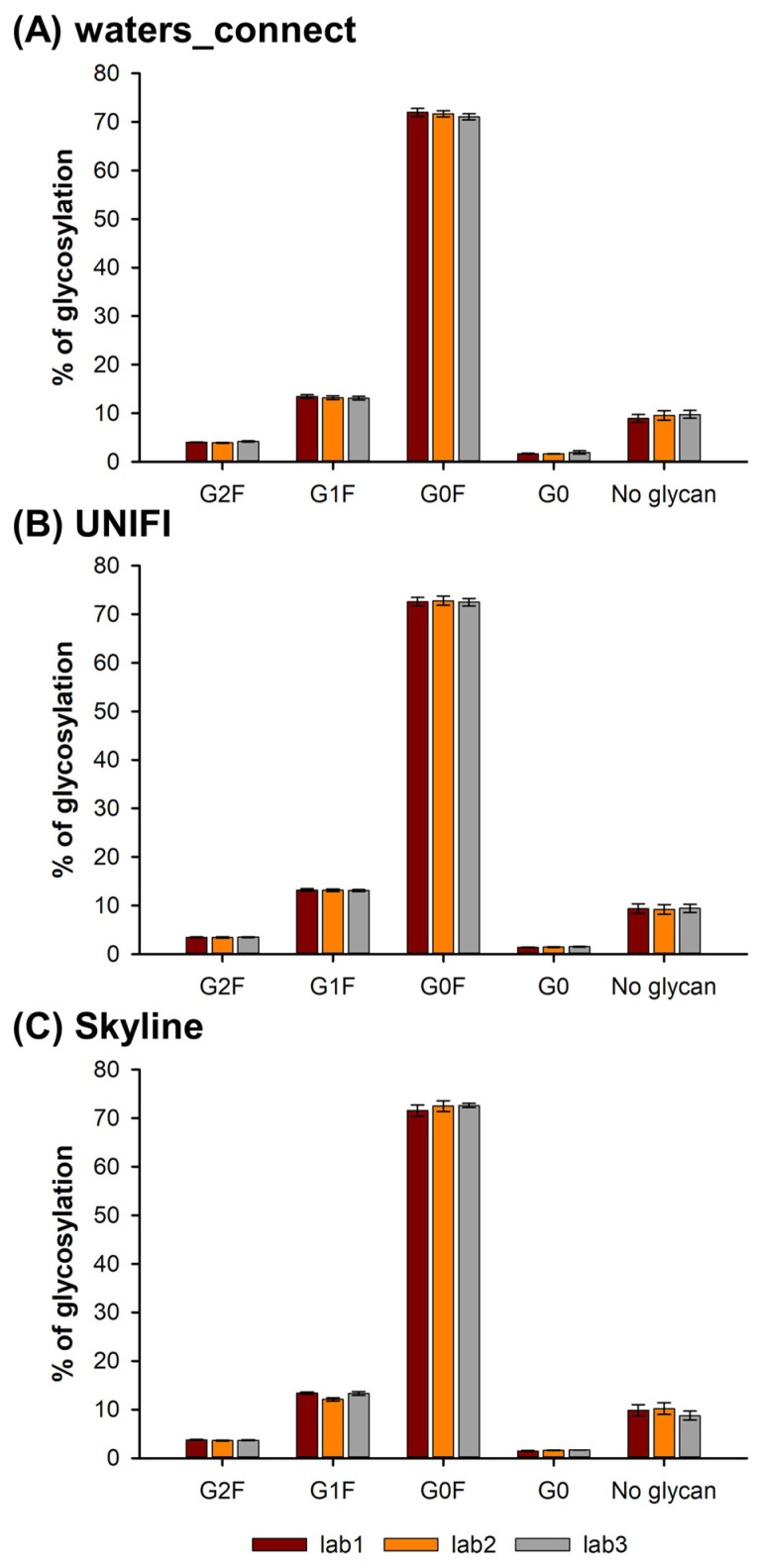
Amounts of G0, G0F, G1F, G2F and non-glycosylated species observed on the TKPREEQFNSTYRVVSVLTVLHQDWLNGK peptide by the three different laboratories (lab1, lab2 and lab3), and by using three different software solutions for data treatment, namely (**A**) waters_connect, (**B**) UNIFI and (**C**) Skyline.

## Data Availability

Not applicable.

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
