# Peer review of "Interlaboratory Evaluation of a User-Friendly Benchtop Mass Spectrometer for Multiple-Attribute Monitoring Studies of a Monoclonal Antibody"

_molecules, 2023, doi:10.3390/molecules28062855_

Round 1

Reviewer 1 Report

Specific comments concerning the manuscript entitled “Interlaboratory evaluation of a user-friendly benchtop mass spectrometer for multiple-attribute monitoring studies of a monoclonal antibody” under reference Molecules-2258878.

The manuscript is well-constructed, clear, documented by many appropriate and clear data and above all, very interesting. This work, conducted by expert teams of research, deserves to be published to enrich the literature on this topic. However, there are few points that deserve to be improved or discussed before its publication in “Molecules”:

A major strength of this study is the conduct of experiments using the same sample and the fact that these experiments were carried out in the three different laboratories with the strictly similar LC-MS equipment. It shows an impressively low, quasi insignificant, variability of the results between the 3 laboratories, which is a good point for the concept of multi-attribute method (MAM) by mass spectrometry. It can be a little more critical for the Data analysis between Waters connect, UNIFI and Skyline. What does it mean? Does it mean that the scientific community using MAM should standardize its material and software in the future? Will we not be creating a manufacturer monopoly (here Waters but it could another instrument provider at the end) for MAM users? Market monopoly does not mix well with science and tend to slowed down its development. I think this issue deserves few lines in the discussion opening in experiment series, this time comparing different LC-MS devices.

Minor issues

1)  Please define in the text the following abbreviation when they appear for the first time. For example, CQA is defined in the abstract but not in the main manuscript (line 54). Even if some abbreviations are obvious for experts, some are less common for ordinary readers like IEX, icIEF, rCE-SDS…

2)  Supplementary data: Figure S2, even if obvious, please indicate the time unit either in the cation or in the y-axis of the graphs.

Author Response

Dear Reviewer,

Thanks for your review and comments. Here is the answer to your points of concern:

[1] It can be a little more critical for the Data analysis between Waters connect, UNIFI and Skyline. What does it mean? Does it mean that the scientific community using MAM should standardize its material and software in the future? Will we not be creating a manufacturer monopoly (here Waters but it could another instrument provider at the end) for MAM users? Market monopoly does not mix well with science and tend to slowed down its development. I think this issue deserves few lines in the discussion opening in experiment series, this time comparing different LC-MS devices

We understand the reviewer concern to relay on a manufacturer monopoly but in the case of MAM the risk is clearly mitigated by the availability of multiple MAM solution on the market (Agilent, Bruker, Protein Metrics, Sciex, Shimadsu, Thermo, Waters), see the list of Mass Spectrometer/Software vendors member of the MAM consortium (www.mamconsortium.org). Moreover, text was added in the discussion section to address this comment from the reviewer, lines 495-502: “The influence of the processing algorithm on the results obtained is a point of attention for the implementation of the MAM approach in the biopharmaceutical industry: if different laboratories are to be involved in a project, several software solutions may have to be used. In this case, the comparability of the results obtained should be evaluated early in the method development process, and if applicable, the method should be validated using the different software solutions in parallel. The use of a well-characterized reference standard such as NISTmAb [35] could also be used to facilitate the comparison across software.”

[2] Please define in the text the following abbreviation when they appear for the first time. For example, CQA is defined in the abstract but not in the main manuscript (line 54). Even if some abbreviations are obvious for experts, some are less common for ordinary readers like IEX, icIEF, rCE-SDS…

To answer this remark, the abbreviations were explained in the text. Lines 44-50 “MAM by LC-MS peptide mapping has the proven capability to replace multiple conventional HPLC / CE-based QC methods such as charge variants (ion-exchange chromatography - IEX, capillary isoelectric focusing - cIEF, capillary zone electrophoresis - CZE) [7], fragments (reduced capillary electrophoresis-sodium dodecyl sulfate - rCE-SDS), oxidation (reversed-phase chromatography - RPC, hydrophobic interaction chromatography - HIC, peptide mapping LC-UV), N-glycans [8,9], sequence variants (LC-MS/MS) and identity tests (ELISA, peptide mapping, LC-UV) [10].”

And lines 56-57: “The aim is also to provide quantitative information on individual site-specific critical quality attributes (CQAs)”.

[3] Supplementary data: Figure S2, even if obvious, please indicate the time unit either in the cation or in the y-axis of the graphs.

The time unit, which is minutes, was added in the figure S2, on the Y-axis.

Reviewer 2 Report

Dear Authors,

the presented manuscript describes the interlaboratory study of the monoclonal antibody - nivolumab using the LCMS technique. I believe the research methodology presented is correct, the article reads smoothly, and the graphic layout is careful. The repeatability of analyzes and convergence of obtained results is a very important aspect of research, especially for difficult compounds such as proteins. For this reason, the comparison of such LCMS analyses, which is on the one hand the technique of choice for this type of research, and on the other hand a very effective tool, in inter-laboratory conditions, using different software, is a valuable scientific contribution. Therefore, I recommend this manuscript after re-checking for editorial errors for publication. Please pay special attention to errors in the formatting of references. For example, some journal abbreviations contain dots, some do not. Therefore, please pay attention to this and standardize the formatting.

Author Response

Dear Reviewer,

Thanks for your review and comments. Here is the answer to your comment:

I recommend this manuscript after re-checking for editorial errors for publication. Please pay special attention to errors in the formatting of references. For example, some journal abbreviations contain dots, some do not. Therefore, please pay attention to this and standardize the formatting.

The manuscript was checked for editorial errors, and the formatting was standardized for the references.

Best regards,
Arnaud